

# Correlation functions and characteristic lengthscales in flat band superconductors

Maxime Thumin[⋆] and Georges Bouzerar[†]

Université Grenoble Alpes, CNRS, Institut NEEL, F-38042 Grenoble, France

⋆ maxime.thumin@neel.cnrs.fr , † georges.bouzerar@neel.cnrs.fr

## Abstract

The possibility of an unconventional form of high temperature superconductivity in flat band (FB) material does not cease to challenge our understanding of the physics in correlated systems. Here, we calculate the normal and anomalous one-particle correlation functions in various one and two dimensional FB systems and systematically extract the characteristic lengthscales. When the Fermi energy is located in the FB, it is found that the coherence length ($\xi$) is of the order of the lattice spacing and weakly sensitive to the strength of the electron-electron interaction. Recently, it has been argued that in FB compounds $\xi$ could be decomposed into a conventional part of BCS type ($\xi_{BCS}$) and a geometric contribution which characterises the FB eigenstates, the quantum metric ($\langle g \rangle$). However, by calculating the coherence length in two possible ways, our calculations show that $\xi \neq \sqrt{\langle g \rangle}$. This may suggest that the link between QM and coherence length is more complex, and leaves us with the open question: what is the appropriate definition of the coherence length in flat-band systems?



# 1  Introduction

Over the past ten years we are witnessing a rapidly growing interest for the physics in dispersion-less bands [1–8]. In flat band (FB) compounds, because the width of these bands is extremely narrow, the Coulomb energy is left as the unique relevant energy scale. This places naturally these systems in the class of highly correlated materials and opens the access to exotic and unexpected physical phenomena and quantum phases. Undeniably, one of the most striking feature is the possibility of high critical temperature superconductivity (SC) in compounds where the Fermi velocity vanishes [9–18]. In contrast to conventional superconductivity, this unconventional form of superconductivity is of inter-band nature.

In other words, the superfluid weight is controlled by the off-diagonal matrix elements (in terms of band index) of the current operator, and the diagonal contribution (conventional contribution) vanishes or is negligible. The superconductivity in FBs is characterised by a geometrical quantity known as the quantum metric (QM). The QM is connected to the real part of the quantum geometric tensor [19, 20] and its square root measures the minimal spread of the Wannier functions. So far, the unique experimental realisation of such an unusual form of superconductivity is very likely the one that has been observed in twisted bilayer of graphene (Moiré) in the vicinity of magic angles [8, 21–26].

It is well known that in conventional BCS systems where the superconductivity is of intra-band nature [27, 28], the coherence length $\xi_c$ is given by $\xi_{BCS} = \frac{\hbar v_F}{\Delta}$ where $v_F$ and $\Delta$ are respectively the Fermi velocity and superconducting gap or pairing amplitude. We recall that $\xi_c$ measures the size of the Cooper pair in real space. Since, in the BCS regime (weak coupling) the superconductivity gap is exponentially small, $\xi_c$ is often extremely large, hence Cooper pairs are highly overlapping with each other. On the other hand, in the strong coupling regime the Cooper pairs can be assimilated to tightly bound non-overlapping composite bosons which at low temperature leads to the well known Bose Einstein condensation phenomenon (BEC) [29, 30].

A natural question arises: what about the case of FB superconductors? Recently, it has been argued that the coherence length in these systems has two contributions, the first is of conventional type and the other is purely geometric in nature [31, 32]. More precisely, it is claimed that the coherence length can be expressed as $\xi_c = \sqrt{\xi_{BCS}^2 + \langle g \rangle}$ where $\langle g \rangle$ is the average of the QM. Hence, if the band is rigorously flat the first term vanishes. Our purpose is to calculate the normal and anomalous one-particle functions in various one and two dimensional FB systems and systematically extract the characteristic lengthscales. In addition, we discuss our findings in connection with the prediction that the coherence length should reduce to $\sqrt{\langle g \rangle}$ when the Fermi energy is located in the FB. To address these issues, we consider four different systems, three of them are one dimensional and the last one is two dimensional: the stub lattice, the sawtooth chain, the Creutz ladder and the $\chi$−lattice. These models and their respective dispersions (in the non interacting case) are depicted in Fig.1. Notice that the $\chi$-Lattice has been originally introduced in Ref. [33]. However, since no specific name has been attributed to this peculiar model,"$\chi$−lattice" has been chosen. In this system, the

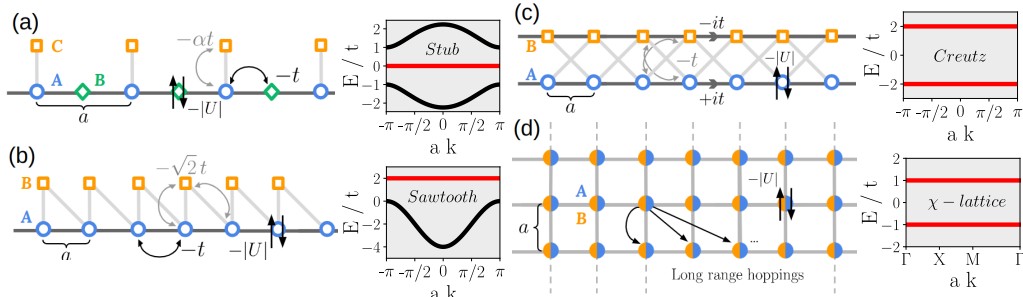

Figure 1: Schematic representation of **(a)** the stub lattice, **(b)** the sawtooth chain, **(c)** the Creutz ladder and **(d)** the two-dimensionnal $\chi$-lattice. Their respective dispersions, in the non interacting case, are depicted in the panels having a grey background. The hoppings and the on-site Hubbard attractive interaction term are depicted in the figure. In the case of the $\chi$-Lattice (two orbitals A and B per site) the hoppings are long range (see main text).

Table 1: Characteristics of the stub lattice, the sawtooth chain, the Creutz ladder and the $\chi$-lattice. 'Tunable QM' means that the system has a degree of freedom that allows the variation of the QM while keeping the FB in the spectrum. 'Uniform paring' means that the pairing is identical for all the orbitals on which the FB eigenstate has a non vanishing weight.

|  | stub | sawtooth | Creutz | $\chi-$Lattice |
|---|---|---|---|---|
| Biparticity | ✓ | ✗ | ✗ | ✓ |
| Tunable QM | ✓ | ✗ | ✗ | ✓ |
| DBs[1] in the spectrum | ✓ | ✓ | ✗ | ✗ |
| Uniform pairing | ✗ | ✗ | ✓ | ✓ |
| Dimensionality | 1D | 1D | 1D | 2D |

range of the extended hoppings is controlled by a single parameter ($\chi$) as it will become more explicit in the next paragraph. The choice of these four different systems is motivated by several intentions. It allows to estimate the impact of (i) the bipartite character of the lattice, (ii) the tunability of the quantum metric, (iii) the absence of dispersive bands in the spectrum, (iv) the lattice dimension, (v) and last the presence of uniform pairings. Each of these five properties, which allow to cover a wide family of systems, has an individual impact on flat-band superconductivity. For this reason, it appears essential to consider various systems to enable a general description of their effects on coherence length. The characteristic features of the different lattices are summarized in Table 1.

## 2 Theory and methods

Electrons are described by the attractive Hubbard model which reads

$$\hat{H} = \sum_{i\lambda,j\eta,\sigma} t_{ij}^{\lambda\eta} \, \hat{c}_{i\lambda,\sigma}^{\dagger} \hat{c}_{j\eta,\sigma} - \mu\hat{N} - |U| \sum_{i\lambda} \hat{n}_{i\lambda,\uparrow}\hat{n}_{i\lambda,\downarrow}, \tag{1}$$

---

[1]Dispersive bands.

where $\hat{c}^{\dagger}_{i\lambda,\sigma}$ creates an electron of spin $\sigma$ at site $\mathbf{r}_{i\lambda}$, $i$ being the cell index and $\lambda$ the orbital index ranging from 1 to $n_{orb}$. $\hat{N} = \sum_{i\lambda,\sigma} \hat{n}_{i\lambda,\sigma}$, $\mu$ is the chemical potential and $|U|$ is the strength of the on-site attractive electron-electron interaction. The hoppings are very short ranged in the stub lattice, the sawtooth chain and the Creutz ladder as depicted in Fig.1. On the other hand, in the $\chi$-Lattice the situation differs, the hoppings are long-ranged, restricted to $(A,B)$-pairs, and given by $t^{AB}_{ij} = -\frac{t}{N_c} \sum_{\mathbf{k}} e^{i\mathbf{k}.\mathbf{r}} e^{i\gamma_{\mathbf{k}}}$ where $\gamma_{\mathbf{k}} = \chi(\cos(k_x a) + \cos(k_y a))$, $\mathbf{r} = \mathbf{r}_j - \mathbf{r}_i$, and $N_c$ being the number of unit cells. The parameter $\chi$ controls both the range of the hoppings and the QM which is given by $\langle g \rangle = \chi^2 a^2/8$ [34].

In this work, we treat the interaction term within the Bogoliubov de Gennes (BdG) approach which consists in the following decoupling scheme,

$$\hat{n}_{i\lambda,\uparrow}\hat{n}_{i\lambda,\downarrow} \stackrel{BdG}{\simeq} \langle \hat{n}_{i\lambda,\downarrow} \rangle \hat{n}_{i\lambda,\uparrow} + \langle \hat{n}_{i\lambda,\uparrow} \rangle \hat{n}_{i\lambda,\downarrow} + \frac{\Delta_{i\lambda}}{|U|}\hat{c}^{\dagger}_{i\lambda,\uparrow}\hat{c}^{\dagger}_{i\lambda,\downarrow} + \frac{\Delta^{*}_{i\lambda}}{|U|}\hat{c}_{i\lambda,\downarrow}\hat{c}_{i\lambda,\uparrow}, \tag{2}$$

where the self-consistent parameters $\langle \hat{n}_{i\lambda,\sigma} \rangle$ and $\Delta_{\lambda} = -|U|\langle \hat{c}_{i\lambda\downarrow}\hat{c}_{i\lambda\uparrow} \rangle$ are respectively the orbital dependent occupations and pairings. $\langle \ldots \rangle$ corresponds to the grand canonical average. Notice, that the total carrier density is defined as $n = N_e/N_c$, where $N_e$ is the total number of electrons, hence $n$ varies from 0 to $2n_{orb}$.

Before we discuss our calculations, we propose to provide some arguments that justify that our approach is meaningful. We first start with the shortcomings. It is well established that the BdG Hamiltonian being quadratic, it is inappropriate to calculate reliably two particles correlation functions (CFs) such as the pairing-pairing correlation function $f_P(\mathbf{r}_i - \mathbf{r}_j) = \langle \hat{\Pi}^{\dagger}_i \hat{\Pi}_j \rangle$ where the on-site pairing operator (s-wave) $\hat{\Pi}^{\dagger}_i = \hat{c}^{\dagger}_{i\uparrow}\hat{c}^{\dagger}_{i\downarrow}$. In the case of the attractive Hubbard model in two dimensional systems, one expects the correlation function $f_P(\mathbf{r})$ to decay algebraically with a $T$-dependent power for $T < T_{BKT}$, and exponentially when $T > T_{BKT}$, where $T_{BKT}$ is the Berezinskii-Kosterlitz-Thouless transition temperature [35–37]. On the other hand, the one-particle CF of the form $f^{\sigma}_{sp}(\mathbf{r}_i - \mathbf{r}_j) = \langle \hat{c}^{\dagger}_{i\sigma}\hat{c}_{j\sigma} \rangle$ always decays exponentially in the superconducting phase. Mean Field theory such as the BdG approach can not describe the change of behaviour of $f_p(\mathbf{r})$ across the BKT transition, since through Wick's theorem two-particles CFs reduce to products of one-particle CFs only. However, in FB systems, one expects the single particle CFs to be well captured within the BdG theory. For instance, it has been shown, that the local occupations, the pairings and the superfluid weight calculated by the numerically unbiased DMRG are in excellent agreement with the mean field values in the Creutz ladder and in the sawtooth chain [12, 38]. It should be emphasised that the agreement found concerns both the weak and the strong coupling regime. In what follows it will be shown that it is as well the case for correlations functions.

To study the characteristic lengthscales in the superconductivity phase at $T = 0$, we define the normal and anomalous CFs,

$$G_{\lambda\eta}(\mathbf{r}) = \langle \hat{c}^{\dagger}_{i\lambda,\sigma}\hat{c}_{j\eta,\sigma} \rangle, \tag{3}$$

$$K_{\lambda\eta}(\mathbf{r}) = \langle \hat{c}_{i\lambda,\uparrow}\hat{c}_{j\eta,\downarrow} \rangle, \tag{4}$$

where the index $i$ (respectively $j$) refers to the unit cell position $\mathbf{r}_i$ (respectively $\mathbf{r}_j$), $\lambda$ (resp. $\eta$) labels the orbitals, and $\mathbf{r} = \mathbf{r}_j - \mathbf{r}_i$. Here, the spin index $\sigma = \uparrow, \downarrow$ is irrelevant, the superconductivity phase being non magnetic. The CF $K_{\lambda\eta}$ is particularly of interest since it allows the extraction of the Cooper pair size. Note that a similar quantity have been used in Ref. [39] to extract the the Cooper pair size in conventional superconductors. Indeed, in the case of a single one dimensional dispersive band problem (conventional SC) it can be shown analytically that $K_{\lambda\lambda}(\mathbf{r}) \simeq \frac{1}{\sqrt{|\mathbf{r}|}}e^{-|\mathbf{r}|/\xi_{BCS}}$ for $|\mathbf{r}| \to \infty$ as addressed in the next paragraph.

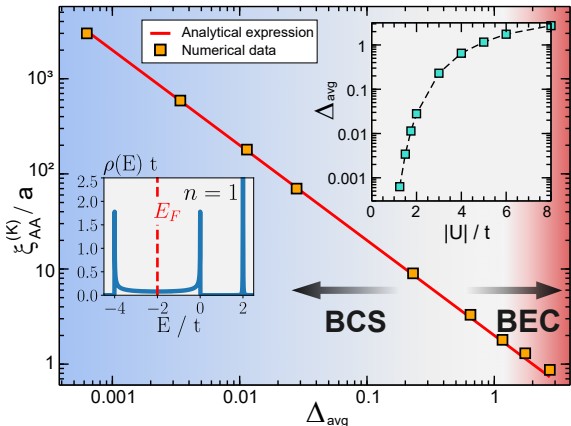

Figure 2: $\xi_{AA}^{(K)}$ as a function of the averaged pairing $\Delta_{avg}$ in the quarter filled sawtooth chain. The red thick line is the BCS formula $\frac{\hbar v_F}{\Delta_{avg}}$ where $\hbar v_F = 2at$. The first inset (top right) shows the correspondence between $|U|$ and $\Delta_{avg}$ and the other one illustrates the density of states for $|U| = 0$, with $E_F = -2t$ for the quarter filling. The BCS regime corresponds to $\xi_{AA}^{(K)} \gg a$ and BEC to $\xi_{AA}^{(K)} \leq a$.

## 3 Results and discussions

### 3.1 Coherence length in dispersive bands

Before we discuss in details the case where the Fermi energy coincides with that of the FB, it is interesting to analyse the situation where it is located inside the dispersive bands. To illustrate this scenario, we consider the quarter filled sawtooth chain. This density corresponds to the half-filling of the lower dispersive band.

In Fig.2, $\xi_{AA}^{(K)}$ is plotted as a function of the averaged pairing $\Delta_{avg}$ in the quarter filled sawtooth chain where $\Delta_{avg} = \frac{1}{2}(\Delta_A + \Delta_B)$ (A and B sites are inequivalent). This characteristic lengthscale is obtained from a fit of the form $\frac{1}{\sqrt{|\mathbf{r}|}} e^{-|\mathbf{r}|/\xi_{AA}^{(K)}}$ of the long distance behaviour of the anomalous CF $K_{AA}(\mathbf{r})$. The BCS-like expression (red thick line in the figure) is defined as $\frac{\hbar v_F}{\Delta_{avg}}$. Here the Fermi velocity $v_F = \frac{2at}{\hbar}\sin(k_F a)$ where $k_F a = \frac{\pi}{2}$ for the quarter filled sawtooth chain. It is striking to see that the excellent agreement found between the numerical data and the BCS expression is not restricted to the weak coupling regime ($\Delta_{avg} \ll t$). Indeed, remarkably the agreement is obtained for values of the average pairing that varies over four decades (see inset of Fig.2), which corresponds to $|U|/t$ that varies from 1 to 8. However, one already observes small deviation from the BCS expression when $|U|/t \geq 5$. We have checked that as $|U|$ increases further the deviation becomes even more pronounced. Numerically, it is found that when $|U|/t \geq 10$, $\xi_{AA}^{(K)} \propto \frac{1}{\sqrt{\Delta}}$ which confirms the existence of a cross-over between BCS and BEC regimes.

### 3.2 The case of half-filled bipartite lattices

We consider the specific case of half-filled bipartite lattices where the number of orbitals in one sublattice is larger than that of the other, implying that at least one FB is located at $E = 0$. We propose to demonstrate the following remarkable property, valid for any $|U|$,

$$G_{\lambda\lambda}(\mathbf{r}) = \frac{1}{2}\delta(\mathbf{r}). \tag{5}$$

In a recent study [40] it has been shown that the Bogoliubov quasi-particle (QP) eigenstates present an interesting symmetry in half-filled systems. If $\mathcal{A}$ (resp. $\mathcal{B}$) denotes the first (resp. second) sublattice which contain $\Lambda_{\mathcal{A}}$ (resp. $\Lambda_{\mathcal{B}}$) orbitals per unit cell, the QP eigenstates can be subdivided in two families $\mathcal{S}^+$ and $\mathcal{S}^-$ defined in what follows. First, a generic QP eigenstate (in momentum space) has the form $|\Psi\rangle = (|\Psi^{\uparrow}\rangle, |\Psi^{\downarrow}\rangle)^t$ where the first $\Lambda_{\mathcal{A}}$ (resp. next $\Lambda_{\mathcal{B}}$) rows of $|\Psi^{\sigma}\rangle$ are the components on sublattice $\mathcal{A}$ (resp. $\mathcal{B}$). This eigenstate belongs to the subspace $\mathcal{S}^+$ (resp. $\mathcal{S}^-$) if $|\Psi^{\downarrow}\rangle = \hat{M}|\Psi^{\uparrow}\rangle$ (resp. $|\Psi^{\downarrow}\rangle = -\hat{M}|\Psi^{\uparrow}\rangle$) where the matrix $\hat{M} = \text{diag}(\hat{1}_{\Lambda_{\mathcal{A}}}, -\hat{1}_{\Lambda_{\mathcal{B}}})$. Additionally, for any finite $|U|$, it has been shown in Ref. [40] that the subset $\mathcal{S}^-$ (respectively $\mathcal{S}^+$) consists exactly in $\Lambda_{\mathcal{B}}$ (respectively $\Lambda_{\mathcal{A}}$) eigenstates of positive or zero energy and $\Lambda_{\mathcal{A}}$ (respectively $\Lambda_{\mathcal{B}}$) eigenstates of strictly negative energy.

Now, start with the definition $G_{\lambda\lambda}(\mathbf{r}) = \frac{1}{N_c}\sum_{\mathbf{k}} e^{i\mathbf{k}.\mathbf{r}}\langle \hat{O}_{\lambda\mathbf{k},\uparrow}\rangle$, where $\hat{O}_{\lambda\mathbf{k},\uparrow} = \hat{c}^{\dagger}_{\mathbf{k}\lambda,\uparrow}\hat{c}_{\mathbf{k}\lambda,\uparrow}$. At $T = 0$, its grand canonical average is given by,

$$\langle \hat{O}_{\lambda\mathbf{k},\uparrow}\rangle = \sum_m \langle \Psi^<_{m\mathbf{k}}|\hat{O}_{\lambda\mathbf{k},\uparrow}|\Psi^<_{m\mathbf{k}}\rangle , \tag{6}$$

where $|\Psi^<_{m\mathbf{k}}\rangle$ are the QP eigenstates of the BdG Hamiltonian of negative energy, $m$ being band index. Using the closure relation, $\sum_{m,s=<,>} |\Psi^s_{m\mathbf{k}}\rangle\langle \Psi^s_{m\mathbf{k}}| = 1$, where the sum runs over QP eigenstates with positive ($s =>$) and negative energy ($s =<$) and the symmetry mentioned above one can show that,

$$\sum_m \langle \Psi^<_{m\mathbf{k}}|\hat{O}_{\lambda\mathbf{k},\uparrow}|\Psi^<_{m\mathbf{k}}\rangle = \sum_m \langle \Psi^>_{m\mathbf{k}}|\hat{O}_{\lambda\mathbf{k},\uparrow}|\Psi^>_{m\mathbf{k}}\rangle , \tag{7}$$

which combined with Eq. 6 leads to $\langle \hat{O}_{\lambda\mathbf{k},\uparrow}\rangle = \frac{1}{2}$ and demonstrates Eq. 5.

It is interesting to remark that our proof can be straightforwardly extended to the case of disordered systems that preserve the bipartite character of the lattice, such as the presence of vacancies or bond disorder.

## 3.3 The stub lattice

The stub lattice is bipartite and offers the possibility to tune the QM without changing the nature of the compact localized eigenstates. The QM is controlled by the A-C hopping $(\alpha t)$ (see Fig.1) and given by $\langle g\rangle = \frac{1}{2|\alpha|\sqrt{4+\alpha^2}}$ [41]. The stub lattice has been studied in great details in Refs. [18, 42]. Here, we restrict our study to the case $\alpha = 0.5$ and $n = 3$ which corresponds to a half-filled FB with $\langle g\rangle \simeq 0.49$.

First, one can already conclude from the previous section that the conventional CFs $(G_{\lambda\lambda})$ are given by Eq. 5, which is indeed what we find numerically for any $|U|$ and any $\alpha$. Figure 3(a) depicts the anomalous CF $K_{CC}$ as a function of $|\mathbf{r}|$ for several values of $|U|$ which correspond to weak, intermediate and strong coupling regime. As it can be clearly seen, in all cases this CF decays exponentially with a lengthscale $\xi^{(K)}_{CC}$ (Cooper pair size) that reduces rapidly as $|U|$ increases. The variation of the extracted lengthscale $\xi^{(K)}_{CC}$ is plotted as a function of $|U|/t$ in Fig.3(b).

In the limit of vanishing $|U|/t$ it is approximately (for this value of $\alpha$) $2a$, then it increases and reaches a maximum for $|U|/t = 1.5$ and beyond it decreases continuously. There is no simple explanation for the origin of this maximum, since for larger values of $\alpha$ it disappears. The inset represents, its behaviour in the large $|U|/t$ limit. It is found that $\xi^{(K)}_{CC} \to 0.125\,a$. As it can be seen, $\xi^{(K)}_{CC}$ crosses $\sqrt{\langle g\rangle} = 0.7\,a$ at $|U|/t \approx 4$ and converges to a much smaller value. The large $|U|/t$ behaviour, is consistent with the fact that in the BEC regime, the Cooper pair size is expected to be very small. Remark that $K_{BB}$ and $K_{AA}$ vary similarly with the same lengthscale.

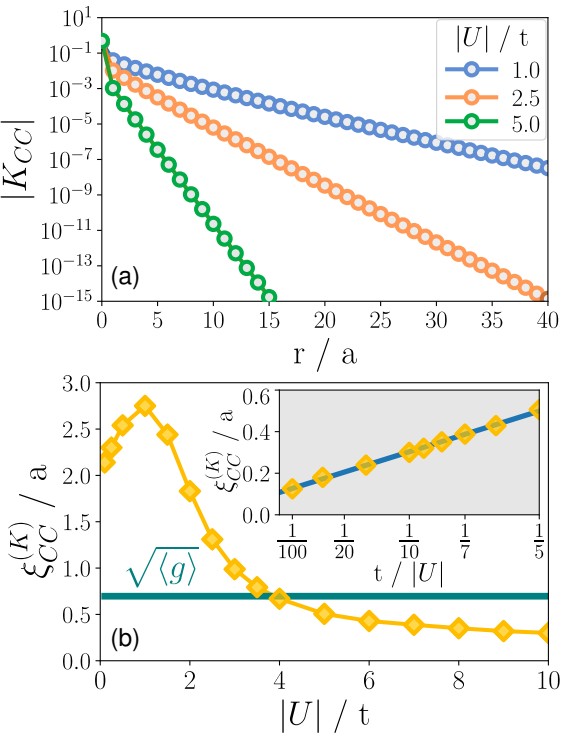

Figure 3: **(a)** $K_{CC}$ as a function of $r$ in the stub lattice for several values of $|U|/t$ (1, 2.5 and 5). **(b)** $\xi_{CC}^{(K)}$ as a function of $|U|$. The (dark-green) horizontal line depicts the square root of the quantum metric $\langle g \rangle$. The inset shows $\xi_{CC}^{(K)}$ for $|U| \gg t$. Here, $\alpha$ is set to 0.5 (see Fig.1) and the carrier density is fixed to $n = 3$ which corresponds to half-filling.

## 3.4 The sawtooth chain

In contrast to the stub lattice, the sawtooth chain as illustrated in Fig.1(b), is a non bipartite lattice and does not allow the tuning of the QM. The FB exists only when the AB-hoppings (1st and 2nd neighbours) are $-\sqrt{2}t$. The superconductivity in the sawtooth chain has been addressed in details in Ref. [38] using a numerically exact method: the DMRG. It has been shown that the BdG approach reproduces accurately the exact results, for both the pairings and the superfluid weight. In Fig.4(a), both $G_{AA}$ and $K_{AA}$ are plotted as a function of $|\mathbf{r}|$, for different values of $|U|$. Here, the electron density is set to $n = 3$ which corresponds to the half-filled FB. As it can be seen, the lengthscales associated to the decay of $G_{AA}$ and $K_{AA}$ are almost identical both in the weak and strong coupling regime. Additionally, the slope appears to vary weakly. Notice that $G_{BB}$ and $K_{BB}$ behave similarly. Fig.4(b) depicts the variation of $\xi_{AA}^{(K)}$ as a function of $t/|U|$. The inset describes the weak coupling regime. In this regime, $\xi_{AA}^{(K)} \approx 0.735\,a$ and almost insensitive to $|U|$. As $|U|$ increases further, $\xi_{AA}^{(K)}$ decays monotonously. As seen in the case of the stub lattice, $\xi_{AA}^{(K)}$ crosses $\sqrt{\langle g \rangle}$ when $t/|U| \approx 0.05$ and converges towards $0.2\,a$. In the sawtooth chain, it can be shown that the minimal QM is $\langle g \rangle = \frac{1}{4\sqrt{3}}$. We should mention as well that our values of $G_{AA}(r)$ are consistent with the DMRG calculations of Ref. [38].

## 3.5 The Creutz ladder

The Creutz ladder depicted in Fig1(c) is particularly interesting since its dispersion consists only in FBs, located at $E = \pm 2t$ in the non-interacting case. As a consequence of the uniform pairings, these bands remain flat when $|U|$ is non-zero. The superconductivity in the Creutz

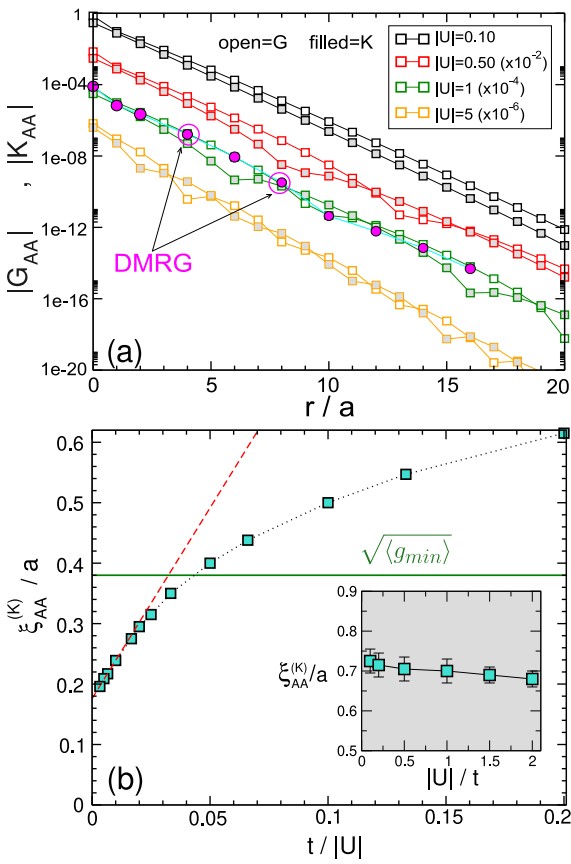

Figure 4: **(a)** $|G_{AA}|$ and $|K_{AA}|$ as a function of $r$ in the sawtooth chain for several values of $|U|$. For the sake of clarity, $|G_{AA}|$ and $|K_{AA}|$ have been multiplied by $10^{-2}, 10^{-4}$ and $10^{-6}$ for $|U| = 0.5$, 1 and 5 respectively. The carrier density is $n = 3$ (half-filled FB). DMRG data for $|U| = 1$ from Ref. [38] are shown as well. **(b)** $\xi_{AA}^{(K)}$ as a function of $t/|U|$. The horizontal lines depicts the square root of the minimal quantum metric $\langle g_{min} \rangle$. The inset represents $\xi_{AA}^{(K)}$ as a function of $|U|$ for small values of $|U|$. The dashed red line is a linear fit for $\frac{t}{|U|} \leq 0.03$.

ladder have been addressed exactly, within the DMRG approach in Refs [12,38]. As in the case of the sawtooth chain, it has been revealed that pairings and superfluid weight are accurately captured by the BdG theory. The A and B sites being equivalent, we focus our attention on $|K_{AA}|$ and $|G_{AA}|$. In addition, we consider the case of the quarter filled ladder (half-filled lower FB) which corresponds to $n = 1$. Both CFs are plotted in Fig.5 as a function of $|\mathbf{r}|$ for several values of $|U|$ ranging from weak to strong coupling regime. As it can be seen these two CFs behave similarly. It is found that there are only two non-vanishing values corresponding respectively to $|\mathbf{r}| = 0$ and $a$. For larger distances, $|K_{AA}|$ and $|G_{AA}|$ are zero within the numerical accuracy. This is illustrated in the inset of Fig.5(b) where for $|\mathbf{r}| = 2a$ the CF $|G_{AA}|$ drops by 16 orders of magnitude. It is found as well that $|K_{AA}|(|\mathbf{r}| = a)$ decays very rapidly as $|U| \geq 1$ and eventually vanishes when $|U| \to \infty$. Thus, the Cooper pair size varies between 1 and 0 where 0 corresponds to $|U| = \infty$. This is consistent with Ref. [43] where the authors have shown that the single-particle propagator vanishes beyond a finite range.

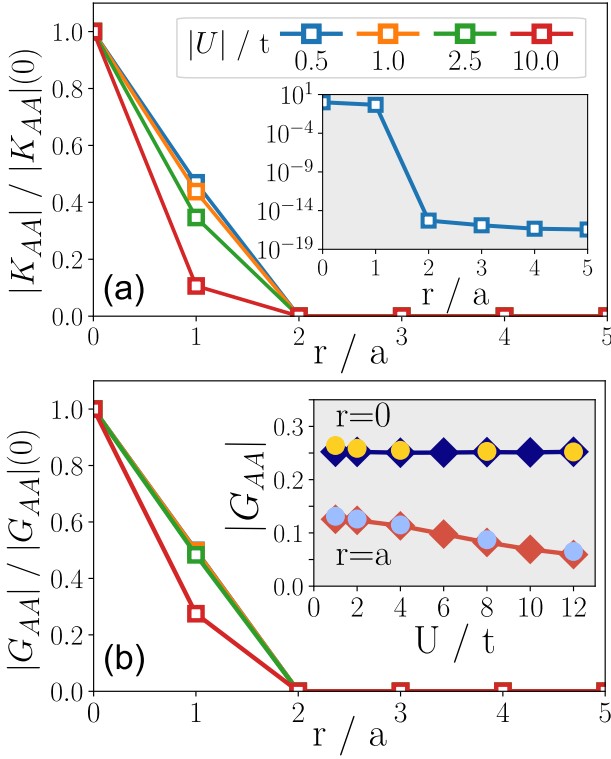

Figure 5: **(a)** $|K_{AA}|$ and **(b)** $|G_{AA}|$, rescaled by their value at $r = 0$, as a function of $r$ in the Creutz ladder for several values of $|U|$. The charge density is fixed $n = 1$. For $r \geq 2a$, both $|K_{AA}|$ and $|G_{AA}|$ are zero within our numerical precision. The inset in (a) represents $|K_{AA}|$ in log scale. The inset in (b) shows $|G_{AA}|$ as a function of $|U|$ for $r = 0$ and $r = a$. Diamonds are our calculations and circles are the DMRG data of Ref. [12].

In the Appendix A, we demonstrate analytically in the case of weak coupling that the CFs are given by,

$$
\begin{aligned}
G_{AA}(r) &= K_{AA}(r) = \frac{1}{4}\delta_{r,0} - \frac{i}{8}\delta_{r,a} + \frac{i}{8}\delta_{r,-a}\,, \\
G_{AB}(r) &= K_{AB}(r) = \frac{1}{8}(\delta_{r,a} + \delta_{r,-a})\,.
\end{aligned}
\tag{8}
$$

We point out the fact that the analytic expression found for $G_{AA}(r)$ is consistent with the exact results obtained from DMRG calculations [12]. Indeed, it has been found (see Fig.10 in this manuscript) that for $r \geq 2a$, $G_{AA} \leq 10^{-12}$.

## 3.6 The $\chi$-lattice

The $\chi$-Lattice is a two dimensional system in which both electronic bands are dispersion-less and located at $E = \pm t$. As mentioned earlier this system has been introduced originally in Ref. [33]. The superconductivity has been addressed within the Quantum Monte Carlo method in Ref. [34] and within a mean field approach in Ref. [32]. We recall that the dimensionless parameter $\chi$ controls both the range of the hoppings and the value of the QM. Here, we focus on the quarter filled system which corresponds to a charge density $n = 1$. As it is the case in the Creutz ladder, the orbitals A and B are equivalent, pairings are identical on both sites. In addition, because the long range hoppings connects A to B sites only, this lattice is bipartite as well.

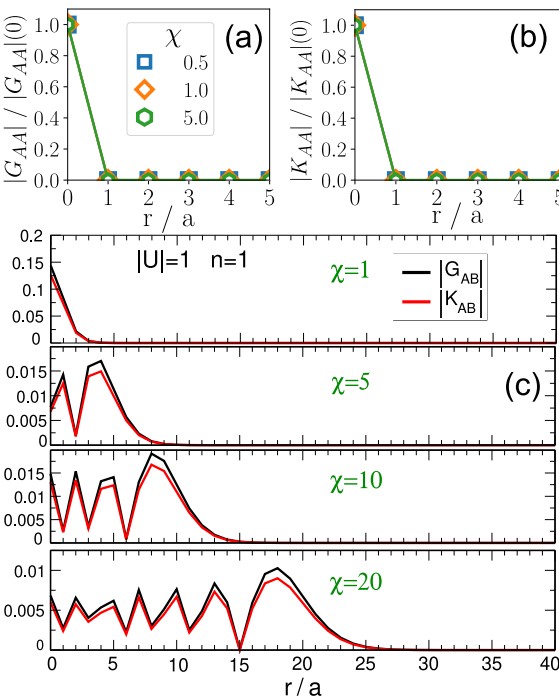

Figure 6: **(a)** and **(b)** $|G_{AA}|$ and $|K_{AA}|$ as a function of $r$ (along the $x-$direction) in the $\chi$-Lattice for several values of $\chi$. **(c)** same as in (a) and (b) for the off-diagonal correlation functions $|G_{AB}|$ and $|K_{AB}|$. The carrier density is $n = 1$ and the Hubbard parameter $|U| = 1$.

Let us now discuss our results. First, for any value of both $|U|$ and $\chi$, with high numerical accuracy we find,

$$\frac{4}{n}G_{\lambda\lambda}(\mathbf{r}) = \frac{|U|}{\Delta}K_{\lambda\lambda}(\mathbf{r}) = \delta(\mathbf{r}), \tag{9}$$

where $\lambda = A, B$. These features are illustrated in Fig.6 (a) and (b). It should be emphasised that the property given in Eq. 5 concerns only the case of half-filled bipartite lattices. Here, our system is quarter filled, which means that our findings are specific to the $\chi$-Lattice. As a consequence, for any $|U|$ the Cooper pair size is zero. In the Appendix B, we have demonstrated analytically Eq. 9 in the weak coupling regime.

More strikingly, we have found that the off-diagonal correlation functions $|G_{AB}|$ and $|K_{AB}|$ exhibit an unexpected behaviour as it can be clearly seen in Fig.6(c). First, one finds that $|G_{AB}|$ and $|K_{AB}|$ are very similar for any value of $\chi$. Furthermore, for a given $\chi$, one can distinguish two distinct regimes. First, for $|\mathbf{r}| \leq \chi a$ the CFs oscillates as $|\mathbf{r}|$ increases. Secondly, when $|\mathbf{r}| \geq \chi a$ it decays monotonously as the distance increases. However, any attempt to fit the tail by a function of the form $r^{-b}e^{-|\mathbf{r}|/c}$ is unsuccessful. Hence, one cannot extract any characteristic lengthscale from these off-diagonal correlation functions. In the Appendix B, we have calculated analytically the $G_{AB}$ and $K_{AB}$ as a function of $\mathbf{r}$ in the limit of small values $|U|$. It is shown that $G_{AB}(\mathbf{r}) = K_{AB}(\mathbf{r}) = \frac{1}{4N_c}\sum_{\mathbf{k}}e^{i\mathbf{k}.\mathbf{r}}e^{-i\gamma_{\mathbf{k}}}$. This means that for this specific lattice the off-diagonal CFs coincides up to a coefficient with the (A,B) hoppings in real-space. In addition, in the limit of large $|\mathbf{r}|$ along the $x$-direction, it is shown that,

$$G_{AB}(\mathbf{r}) \propto (-i)^{n_x}J_0(\chi)\frac{1}{\sqrt{2\pi n_x}}e^{n_x.ln(\frac{e\chi}{2n_x})}, \tag{10}$$

where $\mathbf{r} = (n_x a, 0)$ and $J_0$ is the Bessel function of the first kind and order 0. This clarifies why we could not extract a typical lengthscale from the numerical data plotted in Fig.6(c).

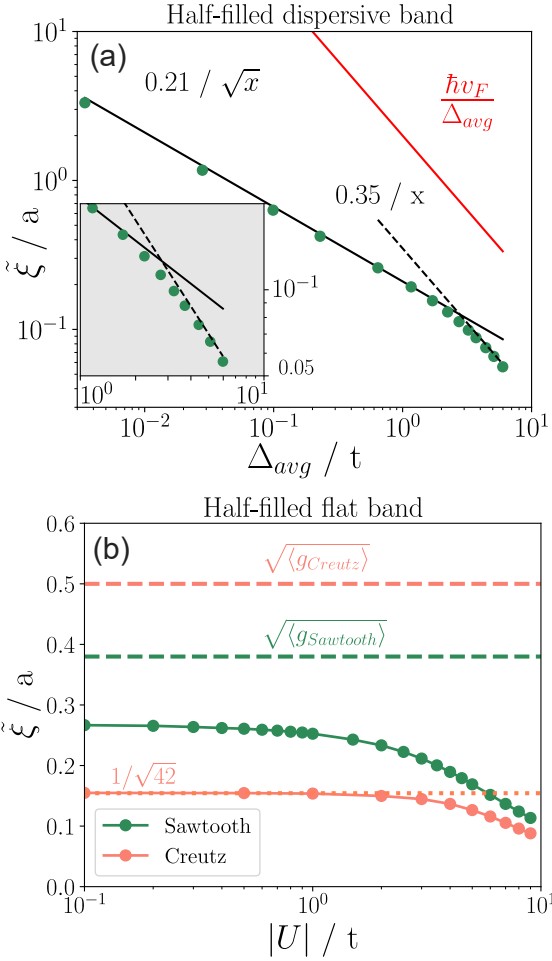

Figure 7: **(a)** $\tilde{\xi}$ (green filled circles) in the sawtooth chain at $n = 1$ (half-filled dispersive band) as a function of the averaged pairing $\Delta_{avg}/t$. Black lines (continuous and dashed) are fits and the red line corresponds to the BCS analytical formula $\xi_{BCS} = \hbar v_F / \Delta_{avg}$. The inset magnifies the region of the cross-over. **(b)** $\tilde{\xi}$ as a function of $|U|/t$ in the Creutz ladder at $n = 1$ and sawtooth chain at $n = 3$ (half-filled flat band). Dashed lines represent $\sqrt{\langle g \rangle}$, $\langle g \rangle$ being the corresponding quantum metric. The dotted line corresponds to the low-$|U|$ limit $\tilde{\xi} = 1/\sqrt{42}$ in the Creutz ladder (see text).

## 3.7 Connection with recent studies

In recent studies [31, 32], it is claimed that the coherence length in quasi FBs can be expressed as, $\tilde{\xi} = \sqrt{\xi_{BCS}^2 + \langle g \rangle}$ where $\langle g \rangle$ is the average of the quantum metric (minimal). The BCS contribution vanishes when the band is rigorously flat. In this paragraph, we discuss the connection between our findings and these recent studies. As done in Ref. [32] we define

$$\xi^2 = -\frac{1}{2\mathcal{M}(0)} \frac{d^2 \mathcal{M}(\mathbf{q})}{dq^2}\Big|_{\mathbf{q}=0}, \tag{11}$$

where the pair correlation function, $\mathcal{M}(\mathbf{q}) = \sum_{ij,\lambda} e^{-i\mathbf{q}(\mathbf{r}_i - \mathbf{r}_j)} \langle \hat{c}_{i\lambda\downarrow} \hat{c}_{i\lambda\uparrow} \hat{c}^\dagger_{j\lambda\uparrow} \hat{c}^\dagger_{j\lambda\downarrow} \rangle$. Within the BdG formalism this leads to

$$\mathcal{M}(\mathbf{q}) = \sum_\lambda |K_{\lambda\lambda}(0)|^2 + 1 - 2G_{\lambda\lambda}(0) + \sum_{r,\lambda} e^{-i\mathbf{q}\mathbf{r}} |G_{\lambda\lambda}(\mathbf{r})|^2, \tag{12}$$

and hence

$$\frac{d^2\mathcal{M}(\mathbf{q})}{dq^2}\bigg|_{\mathbf{q}=0} = -\sum_{r,\lambda}|\mathbf{r}|^2|G_{\lambda\lambda}(\mathbf{r})|^2. \tag{13}$$

Thus, $\tilde{\xi}$ is related to the decay of the normal correlation function $G_{\lambda\lambda}(\mathbf{r})$. From Eq. (5), we conclude straightforwardly that for any bipartite lattice with a different number of orbitals in each sublattice (stub lattice, Lieb lattice...), $\tilde{\xi} = 0$. This applies as well to the $\chi-$Lattice (see Eq (9)). However, in the case of the sawtooth chain and Creutz ladder one cannot conclude, $\tilde{\xi}$ has to be calculated. Figure 7 (a) displays $\tilde{\xi}$ as a function of the averaged pairing at $n = 1$ (half-filled dispersive band) in the sawtooth chain. Fig 7 (b) depicts $\tilde{\xi}$ as a function of $|U|/t$ in the case of half-filled flat band in the sawtooth chain and the Creutz ladder. In (a), $\tilde{\xi}$ appears to scale as $\Delta_{avg}^{-1/2}$ where one would expect $\Delta_{avg}^{-1}$ from a standard BCS analysis. In the appendix C, we provide an analytical justification to this unusual behavior. Furthermore, we emphasize that for a given averaged pairing $\Delta_{avg}$, the values of $\tilde{\xi}$ are found 10 to 100 times smaller than the BCS coherence length. In the case of half-filled FB (panel (b)), we first remark that for both lattices, $\tilde{\xi}$ is much smaller than the lattice parameter, which qualitatively agrees with our previous results. However, in contrast with Ref. [32], these values are clearly smaller than $\sqrt{\langle g \rangle}$ even when $|U| \to 0$. For the Creutz ladder, we note that $\tilde{\xi}$ is finite while it has been found that $\xi^{(K)} = 0$. Using Eq. (8), one can show that $\tilde{\xi} \to a/\sqrt{42} \simeq 0.154\,a$ in the weak coupling regime.

In conclusion, we numerically find that the expression of coherence length given in Ref. [32] does not recover the expected BCS expression in the case of conventional superconductivity. Moreover, when the Fermi energy is located in the flat band, $\tilde{\xi}$ differs from $\sqrt{\langle g \rangle}$ in the weak coupling regime. It even appears that $\sqrt{\langle g \rangle}$ is an upper bound of $\tilde{\xi}$. Finally, it should be emphasized that here, and in contrast to Ref. [32], no projection on the flat band has been done.

## 4 Conclusion

We have investigated the normal and anomalous correlations functions in various flat band systems and extracted the associated characteristic lengthscales. It is found in this study that the size of the Cooper pairs is comparable to the lattice spacing, both in the weak and strong coupling regime. Independently of how extended the hoppings are, it is revealed as well that the normal correlation functions reduce to a Dirac function in the case of half-filled bipartite lattices. In order to clarify a controversial issue regarding the connection between the coherence length and the quantum metric $\langle g \rangle$ we have considered two different definitions of the former. In both cases, it is numerically found that $\xi \neq \sqrt{\langle g \rangle}$ in the weak and strong coupling regimes. The link between quantum metric and coherence length appears controversial, and may require further studies before a clear consensus can be reached. Nevertheless, it is found that $\sqrt{\langle g \rangle}$ provides the correct order of magnitude in the sawtooth chain and for the stub lattice. Finally, we believe as well that this study could motivate new reflections on the concept of coherence length in flat band systems.

## Aknowledgments

We thank George Batrouni, Si Min Chan and Benoit Grémaud for kindly sending us their DMRG data.

# A  The correlation functions in the Creutz ladder

In this appendix we propose to derive analytically the correlations functions G and K as defined in the main text in the quarter filled Creutz ladder. We focus our attention on small values of $|U|$. We restrict our calculation to $T = 0$. The BdG Hamiltonian reads

$$\hat{H}_{BdG} = \sum_k \hat{\Psi}_k^\dagger \begin{pmatrix} \hat{h}_{\mathbf{k}}^\uparrow & \Delta \hat{1}_{2\times 2} \\ \Delta^* \hat{1}_{2\times 2} & -\hat{h}_{-\mathbf{k}}^{\downarrow *} \end{pmatrix} \hat{\Psi}_k \,, \tag{A.1}$$

where we have introduced the Nambu spinor $\hat{\Psi}_k^\dagger = (\hat{c}_{Ak\uparrow}, \hat{c}_{Bk\uparrow}, \hat{c}_{A-k\downarrow}, \hat{c}_{B-k\downarrow})^t$ and the block matrix,

$$\hat{h}_{\mathbf{k}}^\uparrow = \begin{pmatrix} -2t\sin(ka) - \tilde{\mu} & -2t\cos(ka) \\ -2t\cos(ka) & 2t\sin(ka) - \tilde{\mu} \end{pmatrix}, \tag{A.2}$$

where we have introduced $\tilde{\mu} = \mu + \frac{|U|}{4}n$. Because of time reversal symmetry $\hat{h}_{-\mathbf{k}}^{\downarrow *} = \hat{h}_{\mathbf{k}}^\uparrow$. Notice as well that the pairing $\Delta$, uniform because A and B sites are equivalent, can be taken real. Here, the total carrier density $n = 2n_A = 2n_B$ is set to 1.

First, we consider the case $|U| = 0$ for which the chemical potential $\mu = \mu_0 = -2t$. The quasi-particle (QP) eigenvalues are $E_{1,4} = \pm 4t$, and $E_{2,3} = 0$ which is doubly degenerate. The corresponding QP eigenstates are of the form, $|\Psi_i\rangle = (|\psi_i^\uparrow\rangle, |\psi_i^\downarrow\rangle)^t$, where $i = 1, .., 4$.

More precisely they are given by, $|\Psi_1^0\rangle = (0, |\phi_0^+\rangle)^t$, $|\Psi_2^0\rangle = (|\phi_0^-\rangle, 0)^t$, $|\Psi_3^0\rangle = (0, |\phi_0^-\rangle)^t$, and $|\Psi_4^0\rangle = (|\phi_0^+\rangle, 0)^t$, where,

$$|\phi_0^\pm\rangle = \frac{1}{\sqrt{2}} \frac{1}{\sqrt{1 \pm \sin(ka)}} \begin{pmatrix} -\cos(ka) \\ \sin(ka) \pm 1 \end{pmatrix}. \tag{A.3}$$

When the Hubbard term is switched on, we apply a pertubation theory for degenerate pair eigenstates $(|\Psi_2^0\rangle, |\Psi_3^0\rangle)$ that leads to, $E_\pm = \pm\sqrt{(\delta\tilde{\mu})^2 + \Delta^2}$ where $\delta\tilde{\mu} = \tilde{\mu} - \mu_0$. The corresponding QP eigenstates are

$$|\Psi_\pm\rangle = \frac{1}{\sqrt{N^\pm}}\Big(\Delta|\Psi_2^0\rangle + (\delta\tilde{\mu} \pm \sqrt{(\delta\tilde{\mu})^2 + \Delta^2})|\Psi_3^0\rangle\Big), \tag{A.4}$$

where $N^\pm = 2\Big(\delta\tilde{\mu}^2 + \Delta^2 \pm \delta\tilde{\mu}\sqrt{(\delta\tilde{\mu})^2 + \Delta^2}\Big)$.

Using the self-consistent equations for the carrier density which for each spin sector is $1/4$ and the gap equation one finds in the limit of small $|U|$,

$$\delta\tilde{\mu} = 0 + o(|U|^2), \qquad \Delta = \frac{|U|}{4} + o(|U|^2). \tag{A.5}$$

Thus, the QP eigenstates take the simple form $|\Psi_\pm\rangle = \frac{1}{\sqrt{2}}(|\Psi_2^0\rangle \pm |\Psi_3^0\rangle)$, their respective energy being $E_\pm = \mp\frac{1}{4}|U|$.

Using the expressions of $|\phi_0^\pm\rangle$ as given in Eq. A.3, one finds, $\langle c_{Ak,\uparrow}^\dagger c_{Ak,\uparrow}\rangle = \langle c_{Ak,\uparrow}^\dagger c_{Ak,\downarrow}^\dagger\rangle = \frac{1}{4}(1 + \sin(k))$. After a trivial Fourier transform, we finally end up with,

$$G_{AA}(r) = K_{AA}(r) = \frac{1}{4}\delta_{r,0} - \frac{i}{8}\delta_{r,a} + \frac{i}{8}\delta_{r,-a}\,. \tag{A.6}$$

In addition for the off-diagonal CFs it is found that,

$$G_{AB}(r) = K_{AB}(r) = \frac{1}{8}(\delta_{r,a} + \delta_{r,-a})\,. \tag{A.7}$$

These results explain the data plotted in Fig. 5 of the present manuscript. We recall that our proof is restricted to $|U| \leq t$.

# B   The correlation functions in the $\chi$-lattice

In this appendix, our purpose is to derive analytically the correlation functions G and K in the quarter filled $\chi-$Lattice. The BdG calculations are performed for small values of the Hubbard parameter $|U|$ at $T = 0\,K$. The BdG Hamiltonian has the same form as that given in Eq. A.1 of the Appendix A, with $\hat{h}_{\mathbf{k}}^{\uparrow}$ now given by,

$$\hat{h}_{\mathbf{k}}^{\uparrow} = \begin{pmatrix} -\mu - \frac{|U|}{4}n & -te^{-i\gamma_{\mathbf{k}}} \\ -te^{i\gamma_{\mathbf{k}}} & -\mu - \frac{|U|}{4}n \end{pmatrix}, \tag{B.1}$$

where $\gamma_{\mathbf{k}} = \chi(\cos(k_x a) + \cos(k_y a))$.

Notice that the $\chi-$Lattice is both bipartite and time reversal symmetric as well which implies $\hat{h}_{\mathbf{-k}}^{\downarrow *} = \hat{h}_{\mathbf{k}}^{\uparrow}$.

To calculate the QP eigenstates, we use the same notation as those of Appendix A. At $|U| = 0$, the quasi-particle (QP) eigenstates are located at $E_{1,4} = \pm 2\,t$, and $E_{2,3} = 0$ which is doubly degenerate, the chemical potential being $\mu = \mu_0 = -t$. The one particle eigenstates read,

$$|\phi_0^{\pm}\rangle = \frac{1}{\sqrt{2}} \begin{pmatrix} \mp e^{-i\frac{\gamma_{\mathbf{k}}}{2}} \\ e^{i\frac{\gamma_{\mathbf{k}}}{2}} \end{pmatrix}. \tag{B.2}$$

The equations (A.4), (A.5) and (A.6) of Appendix A are valid as well in the case of the $\chi$-Lattice at quarter filling. Thus one straightforwardly gets, $\langle \hat{c}_{A\mathbf{k},\uparrow}^{\dagger} \hat{c}_{A\mathbf{k},\uparrow} \rangle = \frac{1}{4}$, $\langle \hat{c}_{A\mathbf{k},\uparrow}^{\dagger} \hat{c}_{B\mathbf{k},\uparrow} \rangle = \frac{1}{4} e^{-i\gamma_{\mathbf{k}}}$, $\langle \hat{c}_{A\mathbf{k},\uparrow}^{\dagger} \hat{c}_{A-\mathbf{k},\downarrow}^{\dagger} \rangle = \frac{1}{4}$ and $\langle \hat{c}_{A\mathbf{k},\uparrow}^{\dagger} \hat{c}_{B-\mathbf{k},\downarrow}^{\dagger} \rangle = \frac{1}{4} e^{-i\gamma_{\mathbf{k}}}$. It follows that,

$$G_{AA}(\mathbf{r}) = K_{AA}(\mathbf{r}) = \frac{1}{4}\delta_{\mathbf{r},0}, \tag{B.3}$$

and the off-diagonal CFs are

$$G_{AB}(\mathbf{r}) = K_{AB}(\mathbf{r}) = \frac{1}{4}f_{AB}(\mathbf{r}), \tag{B.4}$$

where, we have introduced $f_{AB}(\mathbf{r}) = \frac{1}{N_c}\sum_{\mathbf{k}} e^{i\mathbf{k}\cdot\mathbf{r}}e^{-i\gamma_{\mathbf{k}}}$. Thus, $G_{AB}(\mathbf{r})$ and $K_{AB}(\mathbf{r})$ coincide, up to a constant, with the (A,B) hoppings. We now propose to calculate the analytic expression of $f_{AB}(\mathbf{r})$ for both $|\mathbf{r}|/a \leq \chi$ and $|\mathbf{r}|/a \gg \chi$.

Let us write $\mathbf{r} = (n_x, n_y)$, $f_{AB}(\mathbf{r})$ can be rewritten as the following product,

$$f_{AB}(\mathbf{r}) = I_{n_x}(-i\chi) \cdot I_{n_y}(-i\chi), \tag{B.5}$$

where $I_n(i\chi) = \frac{1}{2\pi}\int_{-\pi}^{+\pi} e^{in\theta} e^{i\chi\cos(\theta)}$ is the modified Bessel function of the first kind and order $n$. We can now rely on the properties of the Bessel functions such as $I_n(-i\chi) = (-i)^n J_n(\chi)$ which leads to

$$f_{AB}(\mathbf{r}) = (-i)^{n_x + n_y} J_{n_x}(\chi) \cdot J_{n_y}(\chi). \tag{B.6}$$

In the regime where $|\mathbf{r}| \leq \chi a$ one can expand the Bessel function [44],

$$J_n(\chi) \simeq \sqrt{\frac{2}{\pi\chi}}\cos\left(\chi - n\frac{\pi}{2} - \frac{\pi}{4}\right), \tag{B.7}$$

and similarly for $J_m(\chi)$. This clearly explains the presence of the oscillations observed in Fig. 6 of the manuscript.

In the opposite limit, more precisely for $\chi \ll \sqrt{|n_x| + |n_y|}$, one has

$$J_n(\chi) \simeq \frac{1}{\Gamma(n+1)}\left(\frac{\chi}{2}\right)^n. \tag{B.8}$$

According to the well known Stirling formula, for $n \gg 1$ one can write $\Gamma(n+1) \simeq \frac{1}{\sqrt{2\pi n}}(\frac{n}{e})^n$. Thus, along the $x-$direction for instance, it implies the following result,

$$f_{AB}(\mathbf{r}) = (-i)^{n_x} \frac{J_0(\chi)}{\sqrt{2\pi n_x}} e^{n_x \ln\left(\frac{e\chi}{2n_x}\right)}. \tag{B.9}$$

This equation explains (i) the rapid decay observed in Fig. 6 of our manuscript and (ii) the impossibility to extract a characteristic lengthscale from the decay at large distance of the off-diagonal correlation functions.

# C   $\tilde{\xi}$ in the half-filled standard one dimensional chain

In this appendix, we provide analytical justifications for the unusual $\Delta$-dependence of $\tilde{\xi}$ observed in Fig. 7. The physics of the sawtooth chain at $n = 1$ being similar to that of a standard half-filled chain, we consider the latter in this appendix. The normal correlation function $G(r)$ as defined in Eq. (4) reads

$$G(r) = \frac{1}{N_c} \sum_k \frac{1}{2}\left(1 - \frac{\varepsilon_k}{E_k}\right) e^{-ikr}, \tag{C.1}$$

where $N_c$ is the number of unit cells, $\varepsilon_k = -2t\cos(ka) - \mu - |U|/2$ is the single particle dispersion, $\Delta$ the superconducting gap, and $E_k = \sqrt{\varepsilon_k^2 + \Delta^2}$ is the quasi-particle energy. We recall that $\mu = -|U|/2$ at half-filling.

We first consider the strong coupling regime ($|U| \gg t$). Equation (C.1) reduces to

$$G(r) = \frac{1}{2}\delta_{r,0} + \frac{t}{2\Delta}(\delta_{r,-a} + \delta_{r,a}), \tag{C.2}$$

from which one immediately gets

$$\sum_r |G(r)|^2 = \frac{1}{4} + \mathcal{O}(\Delta^{-2}), \qquad \sum_r r^2 |G(r)|^2 = \frac{1}{2}\left(\frac{at}{\Delta}\right)^2 + o(\Delta^{-2}). \tag{C.3}$$

Finally, using $\Delta/|U| \to 1/2$ in the strong coupling regime, Eqs. (11), (12), and (13), one finds,

$$\tilde{\xi} = \frac{1}{\sqrt{2}}\frac{at}{\Delta}. \tag{C.4}$$

As can be seen in Fig. 8, the numerical data coincide perfectly with this analytical expression when $\Delta/t \geq 3$ (or equivalently $|U|/t \geq 5$).

Let us now consider the weak coupling regime ($|\Delta| \ll t$). By expanding in Eq. (C.1) $k = \pm\frac{\pi}{2} + q$ one can write

$$G(r) = \frac{1}{2}\left(\delta_{r,0} + I_1(r)\right), \tag{C.5}$$

where

$$I_1(r) = \frac{2t}{\pi}\sin\left(\frac{\pi r}{2}\right)\text{Im}\left\{\int_0^{\frac{\pi}{2}} \frac{\sin(q)}{\sqrt{4t^2\sin^2(q) + \Delta^2}}e^{iqr}dq\right\}. \tag{C.6}$$

This implies that for $r = 2pa$ ($p$ integer) $I_1(r) = 0$. After replacing $\sin(q) \approx q$ one gets

$$I_1(r) = \frac{\Delta}{\pi t}\sin\left(\frac{\pi r}{2}\right)\int_0^\infty \frac{u\sin\left(\frac{\Delta r}{2t}u\right)}{\sqrt{u^2+1}}du. \tag{C.7}$$

Thus, we end up with,

$$I_1(r) = \frac{\Delta}{\pi t}\sin\left(\frac{\pi r}{2}\right)K_1\left(\frac{\Delta r}{2t}\right), \tag{C.8}$$

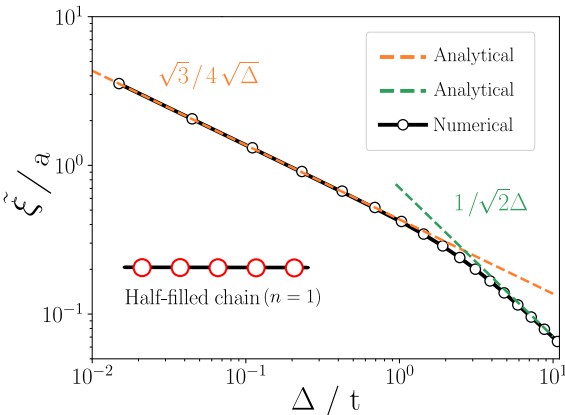

Figure 8: $\tilde{\xi}$ in the half-filled standard chain as a function of the superconducting gap $\Delta$ (open circles). The dashed lines correspond to analytical expressions in the weak and strong coupling regimes.

where we have used $\int_0^\infty \frac{u \sin(\alpha u)}{\sqrt{u^2+1}} du = K_1(\alpha)$, $K_1$ being the first order modified Bessel function of the second kind [45]. Using Eq.(C.5) we can write

$$|G(r)|^2 = \left(\frac{\Delta}{2\pi t}\right)^2 \sin\left(\frac{\pi r}{2}\right)^2 K_1^2\left(\frac{\Delta r}{2t}\right),\tag{C.9}$$

for $r \neq 0$ and $|G(0)|^2 = \frac{1}{4}$. In the limit of vanishing $\Delta$, one numerically finds that $\sum_r |G(r)|^2 = \frac{1}{2}$. On the other hand the $\sum_r r^2 |G(r)|^2$ can be calculated analytically. Indeed, after a change of variable and after replacing the discrete sum by an integral one gets

$$\sum_r r^2 |G(r)|^2 = \frac{2}{\pi^2} \frac{t}{\Delta} \int_0^\infty u^2 K_1^2(u) du = \frac{3}{16} \frac{t}{\Delta},\tag{C.10}$$

where we have used the fact that $\int_0^\infty u^2 K_1^2(u) du = \frac{3}{32}\pi^2$ [45]. Finally, in the coupling regime one finds

$$\tilde{\xi} = \frac{\sqrt{3}}{4}\sqrt{\frac{t}{\Delta}}a.\tag{C.11}$$

As can be seen in Fig.8, over three decades, the agreement between the analytical expression and the full self-consistent numerical calculation is excellent in the weak coupling regime.

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
