# Peer review of "Correlation functions and characteristic lengthscales in flat band superconductors"

_SciPost Physics, doi:SciPost Phys. 18, 025 (2025)_

## Round 1 · Referee Report · Anonymous (Referee 1) · 2024-7-8

Strengths

1- Discussion of essential and timely aspects required for understanding how superconductivity within flatband systems differs from those with dispersive bands 2- Focus on a broad class of prototypical flat band models with key distinct properties 3- By providing a careful derivation of the coherence length the mismatch with existing literature points towards an incomplete understanding of both the physics as well as the current theoretical description

Weaknesses

1- The authors may outline in more depth how the observed discrepancy arises and how this or future work can tackle this issue. 2- The authors may give more context on the challenges and models within the field of flatband superconductivity to help a broader audience understand the raised issue. 3- The authors may improve their figures for a more straight-forward visualization of their key messages.

Report

The authors of the manuscript "Correlation functions and characteristic lengthscales in flat band superconductors" calculate the coherence length for several prototypical flatband models to test the recently published conjecture that the coherence length is determined by the quantum metric, a new characteristic scale of the quantum states significant in the context of flatband physics. They found that within their considered models the coherence length strongly deviates from the quantum metric, which may stimulate new research on the validity of the conjecture, assumptions within their derivations, and the relevance of the quantum metric on the superconductivity in flatband systems in general.

The authors focus on an essential and timely aspect of understanding how superconductivity within flatband systems differs from those in dispersive bands. Whereas the coherence length is well-known as a characteristic quantity, it is still under debate, which quantum state properties determine the unusual properties of flatband superconductivity. The quantum metric is commonly believed as one key quantity here. In addition, it remains an open question to understand the validity and accuracy of various analytical and numerical approaches to tackle the regime where the bandwidth is vanishing.

The presented findings are in line with a growing number of seemingly contradictory results that require a careful analysis of statements that are well-known for "ordinary" superconductors but require revisions in this context. It is of urgent and broad interest to report these contradictions together with a careful analysis, which will guide us toward a better understanding of flatband superconductors. Thus, I agree with the authors's indication that their work opens a new pathway in an existing or a new research direction, with clear potential for multi-pronged follow-up work.

As discussed below in the requested changes, I encourage the authors to improve further their manuscript regarding (1) a clearer description of how to tackle the discrepancy, (2) a broader context of the choices made within their analysis, and (3) a broadly accessible presentation of their work. I believe that these improvements will lift their manuscript to a level where I can easily justify publication within SciPost Physics.

Requested changes

1- I encourage the authors to deepen their discussion of its connection with recent studies. The authors relate the deviation of their results to those within the literature to the decoupling of the Hubbard term and the projection of the operators onto the lowest flatband. Throughout the manuscript, the authors have already justified their approach by relating it to DMRG calculations. It would strengthen their conclusions when the authors

(i) Be more mathematically precise about how the decoupling differs, (ii) Discuss how the projection mechanism may account for the different findings, and (iii) Give arguments, which approach is expected to capture the experimental results, and give, if possible, corresponding references.

I understand that a complete answer is far beyond the scope of this first work so I would be satisfied by a more in-depth description of the problem, related deviations seen in other literature, and a more in-depth guidance for future research.

2- Some parts need some more explanations and context. In particular,

(i) What do the authors precisely mean by "intraband" and "interband nature" in the introduction? (ii) Why should we have doubts about the previous literature on the relation between the quantum metric and coherence length in the first place? (iii) Why are these four models chosen in light of potential deviations? What can the insights of each model contribute to the big picture? (iv) What is the role of the BCS - BEC crossover in light of the investigated deviation between existing and this literature?

3- The presentation requires some minor revisions:

(i) I would ask the authors to reduce the number of abbreviations to a minimum as they are making the document less accessible in particular for those not experienced with the particular subfield. (ii) I encourage the authors to work on the presentation of their figures. Whereas several of them are very clear and easily accessible such as Fig. 1 and 2, others lack a consistent choice of labels, fonts, size, etc. (iii) The authors might check for some typos/double-use of labels. An example of this would be the double use of the index alpha for both the orbital label and the hopping strength.

Recommendation

Ask for minor revision

---

## Round 1 · Referee Report · Sebastiano Peotta (Referee 2) · 2024-7-19

Strengths

1- Clear presentation 2- Sound analytical derivations 3- Interested numerical data 4- Topical and important subject

Weaknesses

1- The limits of validity of previous results, which the author criticize, have not been appropriately included and discussed 2- The interpretation of the numerical results is probably not correct 3- The discussion of the results could be extended considerably

Report

In their work “Correlation functions and characteristic lengthscales in flat band superconductors” the authors analyze the behavior of correlation functions in various lattices in the presence of attactive interactions. Their focus is to understand how the Cooper pair size (also known as the coherence length) depends on band structure invariants such as the quantum metric. A possible relation between quantum metric and coherence length has been proposed in some recent works, specifically Refs. 31, 32, and the aim of the authors is to investigate to what extent this relation holds true.

This work is overall well written and the numerical data and the derivations of the analytical results seem sound in my opinion. On the other hand, there are some clear flaws in the way the results are interpreted leading to the conclusion by the authors that the relation between coherence length and quantum metric is not valid. While I cannot express myself on the validity of this relation, the arguments proposed by the authors are in my opinion not solid enough to conclude that the coherence length is disconnected from the quantum metric.

The first point is that the authors should clearly state the limits of validity of the relation between quantum metric and coherence length. Indeed, I do not expect the relation to hold for arbitrary values of the interaction strength. The benchmark in this sense is the well established result that the superfluid weight is proportional to the integral of the quantum metric over the Brillouin zone. This result is valid only for sufficiently small values of the Hubbard coupling U, which should be no larger the the band gap separating the partially filled flat band from other fully filled or empty bands. I would expect that the same restrictions hold also in the case of the results of Ref. 31, 32 regarding the coherence length, although I have not investigated the matter very carefully. For instance it is shown both in Fig. 3b and Fig. 4b that for large enough values of U the coherence length, extracted from the decay of the anomalous correlation function, becomes smaller than the lower bound given by the quantum metric derived in Ref. 31, 32. This is not surprising in my opinion as this occurs for values of U that are comparable or larger than the band gap (the authors should check this), namely a regime where the coherence length is not controlled by the properties of a single partially filled band but also by other bands that are close in energy. For very larger values of U, a real space picture becomes more appropriate than a momentum space picture since a strong attractive Hubbard interaction leads to the formation of Cooper pairs in which both particles sit with high probability on the same lattice site. In summary, the numerical evidence provided in the manuscript regarding the stub lattice and the sawtooth lattice (Figs. 3 and 4) does not support the claim by the authors that the coherence length is disconnected from the quantum metric since this conclusion is drawn from the data for values of U comparable to or larger than the band gap, that is in a regime where a simple relation between quantum metric and coherence length is a priori not expected. The authors should carefully investigate the limits of validity of the results derived in Refs. 31 and 32 and, based on their findings, reassess their interpretation of the numerical data for the stub and sawtooth lattices.

The Creutz lattice, and also to some extent the χ-lattice, is peculiar since the single-particle correlation funtions (both normal and anomalous) become zero for distances larger than one lattice spacing. First of all, I would like to point out that the vanishing of single-particle correlation functions of the Creutz ladder beyond a finite distance is not a surprising result, as stated by the authors, rather it is a straightforward consequence of the local integrals of motion of the Creutz ladder with an Hubbard interaction, which have been found in [M. Tovmasyan, et al. Phys. Rev. B 98, 134513 (2018)]. Most importantly, the compact character of the correlation functions in the Creutz ladder implies that the coherence length, defined as the rate of the exponential decay of the correlation functions, is always zero regardless of the interaction strength. In my opinion, the Creutz ladder, rather then providing a counterexample to the statement that the coherence length is controlled by the quantum metric in some regime, shows that we should consider some other definition of the coherence length. A more appropriate definition in my opinion is to define the square of the coherence length as the average of the square of the distance between the two particles forming a Cooper pair. This is consistent with the role played by the quantum metric in providing a lower bound to the spread of Wannier functions, which is also defined as the average of the distance squared. Most importantly, this definition would give a nonzero and interaction-dependent value also in the case of the Creutz ladder. It might be that the results of Ref. 31 and 32 regarding the relation between quantum metric and coherence length refer to the latter definition, which is in my opinion more meaningful. The author should carefully check this point before drawing any conclusion.

The case of the χ-lattice also shows that a more appropriate definition of the coherence length would be in terms of the average of the square of the interparticle distance. Indeed, the correlation functions are either zero at large distances if the orbital labels are equal, or they have an anomalous decay behavior if the orbital labels are different. While the definition of the coherence length in terms of the exponential decay length is not very useful in this case, because no good fit can be obtained in the case of different orbitals, the definition based on the distance square is and would probably give meaningful results. I would be interested in see how the coherence length, defined as the average of the distance squared, behaves in the case of the χ-lattice. Note that the average involves also an average over all orbitals, therefore the different behaviors with respect to the orbital choice would not matter. Also, from the fact that the single-particle correlation functions vanish identically at any nonzero distance if the orbital indices are the same have prompted the author to state that “As a consequence, for any |U | the Cooper pair size is zero.” In my opinion this statement is not justified because to evaluate the coherence length one should consider all possible orbital choices.

In the paragraph “Connection with recent studies” the author state that: “Based on the fact that the coherence length is extracted from the long distance decay of Kαα our findings clearly contradicts their prediction.” The fact that the coherence length is extracted from the rate of the exponential decay at large distances is not obvious and the authors should check that this is in fact the same definition used in Refs. 31 and 32. As discussed above, other definitions are possible and probably more meaningful even in special cases such as the Creutz ladder and the χ-lattice. Also, there is no valid reason why the correlation function Kαᵦ should be neglected for α different from β. For these reasons I do not agree with the discussion presented in this paragraph of the manuscript. To remedy this, it would important for the authors to gain a more in depth understanding of the previous results of Ref. 31 and 32 in order to give a better interpretation of their numerical results.

Whereas the manuscript is well written clear, few points require some attention: - In the introduction it is written “The QM is connected to the real part of the quantum geometric tensor [19, 20] and provides a measure of the typical surface associated to the FB Bloch eigenstates”. It is unclear for me what “measure of the typical surface associated to the FB Bloch eigenstates” means. - At the beginning of the Theory and Method section: “On the other hand, the one-particle CF of the form ... always decays exponentially both in the superconducting phase and in the normal phase.” In the absence of a gap, that is in the normal phase, the correlation function does not have an exponential decay behavior in general. - In the section “Coherence length in dispersive bands” it is stated: “This density corresponds to the half-filling of the lower dispersive band.” It would be useful to include a plot of the dispersion of the sawtooth lattice (as well as the stub lattice later on) to help the uninitiated reader understand this statement. - In the conclusion: “It is found that the size of the Cooper pairs is less than one lattice spacing, both in the weak and strong coupling regime.” This is a rather generic statement, which is for sure not applicable to all of the lattices studied in the manuscript. The authors should specify to which lattices they are referring to. - The Conclusion should be significantly expanded in light of the above considerations.

Requested changes

1- Understand in detail previous works claiming that there is a relation between quantum metric and coherence length. In particular check what is the operational definition of the coherence length used there. 2- Estimate the coherence length using using the mean square distance rather than the rate of exponential decay of correlation functions. 3- Reassess the interpretation of the numerical data in light of the considerations given in the report. 4- Address the minor points listed at the end of the report. 5- Add one relevant citation [Tovmasyan et al.]

Recommendation

Ask for major revision

---

## Round 2 · List of Changes

3 minor correction suggested by the referee

---

## Editorial Decision

published